# Synthesis of a Chiral 3,6T22-Zn-MOF with a T-Shaped Bifunctional Pyrazole-Isophthalate Ligand Following the Principles of the Supramolecular Building Layer Approach

**DOI:** 10.3390/molecules27175374

**Published:** 2022-08-23

**Authors:** Dennis Woschko, Simon Millan, Muhammed-Ali Ceyran, Robert Oestreich, Christoph Janiak

**Affiliations:** Institut für Anorganische Chemie und Strukturchemie, Heinrich-Heine-Universität Düsseldorf, 40225 Düsseldorf, Germany

**Keywords:** metal-organic frameworks (MOF), zinc, topologies, T-shaped linker, bifunctional pyrazole-carboxylate linker, conglomerate, enantiomeric excess, chiral space group

## Abstract

The metal–organic framework (MOF) [Zn(Isa-az-tmpz)]·~1–1.5 DMF with the novel T-shaped bifunctional linker 5-(2-(1,3,5-trimethyl-1H-pyrazol-4-yl)azo)isophthalate (Isa-az-tmpz) was obtained as a conglomerate of crystals with varying degrees of enantiomeric excess in the chiral tetragonal space groups P4_3_2_1_2 or P4_1_2_1_2. A topological analysis of the compound resulted in the rare **3,6T22**-topology, deviating from the expected **rtl**-topology, which has been found before in pyrazolate-isophthalate-functionalized MOFs using the supramolecular building layer (SBL) approach. **3,6T22**-[Zn(Isa-az-tmpz)]·~1–1.5 DMF is a potentially porous, three-dimensional structure with DMF molecules included in the corrugated channels along the *a* and *b*-axis of the as synthesized material. The small trigonal cross-section of about 6 × 4 Å (considering the van der Waals surface) prevents the access of N_2_ and Ar under cryogenic conditions. After activation, only smaller H_2_ (at 87 K) and CO_2_ (at 195 K) are allowed for gas uptakes of 2 mmol g^–1^ and 5.4 mmol g^–1^, respectively, in the ultramicroporous material, for which a BET surface area of 496 m^2^·g^–1^ was calculated from CO_2_ adsorption. Thermogravimetric analysis of the compound shows a thermal stability of up to 400 °C.

## 1. Introduction

Metal–organic frameworks (MOFs) are a much-studied topic with a wide variety of potential applications, interesting properties and topologies [1,2,3]. There are many factors that can influence the synthesis, growth and the structure of MOFs. One approach to designing MOFs with a certain structure and underlying net is the ‘supramolecular building layer approach’, formulated by Guillerm et al. [4]. Within this approach, multiple ways were rationalized to obtain MOFs with certain topologies through the inter-connection of two-dimensional nets with accessible perpendicular bridging sites. One of the best-known strategies is the use of 4,4′-bipyridine derivates to connect carboxylate-based paddlewheel clusters along the axial open metal sites, thereby effectively turning the tetragonal 4-c (four-connected) nodes into octahedral 6-c-nodes [5,6,7,8]. Seki et al. were among the first to utilize this strategy using copper (II) terephthalate and triethylenediamine as a pillaring ligand to synthesize a mixed-ligand MOF with a **pcu** topology [9]. The success of this strategy led to the synthesis and use of new bifunctional ligands, wherein the dicarboxylate group constructs a two-dimensional net while a N-heterocycle functions as an axial pillaring unit, unifying the mixed-ligand approach. This strategy has been termed ‘ligand-to-axial pillaring’ by Eubank et al. based on the utilization of T-shaped bifunctional ligands that function as a 3-c node [4,10]. Using this strategy, a plethora of 3,6-c connected MOFs, with topologies such as **apo** (α-PO2), [10] **eea** (based on the Kagomé-lattice (**kgm**)) [10,11,12], **pyr** (pyrite) [12] and **rtl** (rutile) [10,12,13,14], have been synthesized.

Among the utilized bifunctional T-shaped ligands to synthesize these 3,6-c connected MOFs the pyrazole-carboxylate ligands have some especially interesting properties compared to the more commonly used pyridine–carboxylate ligands, since they have an additional NH-function, which can act as an additional interaction site [13,15]. 5-(Pyrazole-4-yl)isophthalic acid (H_3_Isa-pz) (Figure 1), as the simplest T-shaped pyrazole-carboxylate ligand, has been used by Ma et al. to synthesize **rht**-MOF-pyr. This MOF consists of a copper paddlewheel unit connected through the isophthalate oxygen atoms and a trinuclear copper cluster connected through both pyrazole nitrogen atoms [16]. The azo-functionalized T-shaped ligand 5-(4-(3,5-dimethyl-1H-pyrazolyl)azo)isophthalic acid (H_3_Isa-az-dmpz) (Figure 1) has been utilized by Millan et al. to obtain two **rtl**-MOFs with copper and zinc [13]. The copper MOF **rtl**-[Cu(HIsa-az-dmpz)] showed a good CO_2_ uptake while the zinc MOF **rtl**-[Zn(HIsa-az-dmpz)] went through an irreversible transformation into a non-porous compound [13]. The azo group in the linkers H_3_Isa-az-dmpz and H_2_Isa-az-tmpz could be used for trans-cis photoisomerization effects in future studies [17,18,19].

Herein, we present the successful synthesis of a new 3,6-c connected MOF [Zn(Isa-az-tmpz)], following the ligand-to-axial pillaring approach. Instead of the expected **rtl**-topology, which had been found for the MOFs **rtl**-[Cu(HIsa-az-dmpz)] and **rtl**-[Zn(HIsa-az-dmpz)] with the related HIsa-az-dmpz^2–^ linker [13], here, the rare chiral **3,6T22**-topology was obtained in [Zn(Isa-az-tmpz)]·~1–1.5 DMF through the reaction of Zn(NO_3_)_2_·4 H_2_O with the newly synthesized T-shaped pyrazole-carboxylate ligand Isa-az-tmpz^2–^ (5-(2-(1,3,5-trimethyl-1H-pyrazol-4-yl)azo)isophthalate). Chiral MOFs are of interest for chiral separation and catalysis [20,21,22].

## 2. Results and Discussion

The compound [Zn(Isa-az-tmpz)]·~1–1.5 DMF was obtained through the reaction of Zn(NO_3_)_2_·4 H_2_O with 5-(2-(1,3,5-trimethyl-1H-pyrazol-4-yl)azo)isophthalic acid (H_2_Isa-az-tmpz) in DMF at 80 °C. This synthesis yielded yellow octahedral-shaped crystals of formula [Zn(Isa-az-tmpz)]·~1–1.5 DMF. The comparison between the morphology of the single crystal used for the measurement and the theoretical morphology calculated with MERCURY from the crystal structure showed a decent match (Figure 1) [23].

The asymmetric unit consisted of one Zn(II) ion and one fully deprotonated Isa-az-tmpz^2–^ ligand, as well as a strongly disordered DMF molecule which was removed with the SQUEEZE function in PLATON [24,25]. Each Zn(II) ion had a square-pyramidal coordination with Zn-O bond lengths between 2.024(2) and 2.046(2) Å in equatorial positions and a similarly long Zn-N bond of 2.039(2) Å in axial position. Two of these Zn(II) ions form a paddlewheel cluster connected through four isophthalate and two pyrazole units of six different linker molecules to a 3D network (Figure 2a). The isophthalate and pyrazole-ring plane are tilted by about 20° with respect to each other.

Paddlewheel clusters are commonly found in Zn-MOFs, with Zn-HKUST-1 and SDU-1 being two examples [26,27]. These paddlewheel clusters can exhibit symmetries up to D_4h_ depending on the symmetries of the ligand and the molecules or ligands in axial positions. Due to the low symmetry of Isa-az-tmpz^2–^, the clusters in the network [Zn(Isa-az-tmpz)] have a reduced C_2_ symmetry without an inversion center or mirror faces (Figure 2b).

Each paddlewheel cluster was connected to six ligand molecules (Figure 2b), which in turn were linked to two further paddlewheel clusters to form a 3D-network (Figure 2c). This 3D-network can be separated into 2D-chains, consisting of the paddlewheel units interconnected through the isophthalate functionality of the ligand. As a result, these form a left-handed 4_3_ or a right-handed 4_1_ helix along the crystallographic 4_3_ or 4_1_ axis, respectively, which was colinear with the c-axis (Figure 2d).

The compound crystallizes in the chiral, enantiomorphic tetragonal space groups P4_3_2_1_2 or P4_1_2_1_2 as a conglomerate of crystals, with varying degrees of enantiomeric excess. The four investigated crystals (Table 1) were refined as inversion twins with ratios of about 7:3 and 6:4 or 4:6 and 2:8 of both enantiomorphic forms [28,29]. Thus, the individual investigated crystals were not enantiopure (homochiral) but only of enantiomeric excess. The formation of fourfold helices in isophthalate-MOFs was also seen in [Al(OH)(isophthalate)] (CAU-10-H, including benzene-functionalized derivatives) [30,31,32].

A packing analysis of the network [Zn(Isa-az-tmpz)] shows the absence of π-π and C-H···π interactions within the structure [37]. The steric constraints of the three methyl groups on the pyrazolyl ring prevent such π-π interactions between the aromatic rings of the ligand. The network [Zn(Isa-az-tmpz)] follows the basic principles of the ‘ligand–to–axial pillaring’ strategy of the SBL approach consisting of a paddlewheel cluster, which can be described as an octahedral 6-c node and a T-shaped ligand, which works as a trigonal 3-c node. Deviating from the SBL approach, the topological analysis of the structure with the program ToposPro [38,39] and the Topcryst database [40,41] yielded the rare chiral **3,6T22**-topology as the underlying net, instead of the expected **rtl**-topology (Figure 3).

To date, the **3,6T22**-topology has been reported in multiple G⊂Cd(L)_2_ (G = guest molecule; L = 4-amino-3,5-bis(4-pyridyl-3-phenyl)-1,2,4-triazole) host@guest complexes, reported by Liu et al. [42,43]. More recently, the **3,6T22**-topology has been observed in the Cd- and Fe-based MOFs [(Me_2_NH_2_)Cd_3_(OH)(H_2_O)_3_(Tatab)_2_] (Tatab = 4,4′,4″-s-triazin-1,3,5-triyltri-p-amino-benzoate) and [Fe_2_M(Bptc)] (M = Fe, Co, Ni, Zn; Bptc = biphenyl-3,4′,5-tricarboxylate), synthesized by Liu et al. [44] and Wang et al. [45], respectively. The other **3,6T22** MOFs all had a 3c-linker node connected to an octahedrally distorted 6c-metal (SBU) node. In G⊂Cd(L)_2_ the SBU distortion originates from four pyridyl and two triazole donors around the six-coordinated Cd atom (see Appendix A). In [(Me_2_NH_2_)Cd_3_(OH) (H_2_O)_3_(Tatab)_2_] and [Fe_2_M(Bptc)] the trinuclear {M_3_(µ_3_-O)(H_2_O)_3_(O_2_C-)_6_} SBU was a trigonal prism, which also represents a distortion from an octahedron (Appendix A). In [Zn(Isa-az-tmpz)]·~1–1.5 DMF the dinuclear paddlewheel SBU gave rise to a tetragonally distorted, elongated octahedron (Appendix A). In G⊂Cd(L)_2_, [Fe_2_M(Bptc)] and [Zn(Isa-az-tmpz)] the 3c-linker node also was asymmetric, as it had short and long bonds to the SBU (Appendix A). Only the Tatab^3–^ linker was trigonal symmetric (Appendix A). The Bptc^3–^ linker was also a T-shaped linker like Isa-az-tmpz^2–^, albeit with a tricarboxylate donor set. To the best of our knowledge, this was the first work that shows that this topology can be achieved using a T-shaped bifunctional pyrazole–dicarboxylate ligand following the ‘ligand-to-axial pillaring’ approach.

The solvent-depleted 3D network [Zn(Isa-az-tmpz)] has potential porosity from the identical perpendicular corrugated channel systems along the a- and b-axes with trigonal cross-selections of about 6 × 4 Å, which could only accommodate a sphere of about 3 Å diameter (considering the van der Waals surface) (Figure 4). A solvent accessible volume (SAV) of 1661 Å^3^ or 39 vol% out of the unit cell volume of 4223 Å^3^ was calculated with PLATON [24] for the solvent-depleted structure. The SAV of 1661 Å^3^ calculates into a specific pore volume of 0.34 cm^3^ g^–1^ according to (SAV × N_A_)/(Z × M_asym unit_); (N_A_ = Avogadro’s constant: 6.022·10^23^ mol^–1^, Z = number of asymmetric formula units, M_asym unit_ = molecular weight of asymmetric formula unit in g mol^–1^; see Appendix A).

The representative nature of the selected crystal from [Zn(Isa-az-tmpz)]·~1–1.5 DMF and the phase-purity of the bulk material was confirmed by a positive match between the simulated powder X-ray diffraction (PXRD) pattern and experimental pattern of the as-synthesized material (Figure 5a,b). For the prospective gas sorption studies, DMF was exchanged with acetone, and afterwards the material has been dried with supercritical CO_2_. At this stage, no phase change or loss of crystallinity in the bulk material could be observed with PXRD (Figure 5c,d). The solvent exchange does not influence the general structure of the MOF, as the peak positions remained unchanged. However, the difference in electron density in the pores from the solvent exchange can affect the peak intensities [46].

Thermal analysis of the supercritically dried (sc-dried) sample with TGA showed only a slightly decreased mass loss compared to the acetone-washed sample. While in one sample, the solvent could nearly be completely removed according to TGA (Appendix A), in another sample about 15 mass% of DMF up to 250 °C remained in both the acetone-washed and the subsequently sc-dried sample (Appendix A). Hence, the sc-dried material was additionally heated for 3 h at 120 °C under high vacuum to further activate the sample before measurement, which yielded a material with low crystallinity (Figure 5e). This indicates a partial collapse of the 3D network structure during solvent removal under too harsh conditions. Following a volumetric nitrogen sorption experiment at 77 K, no gas uptake could be seen. A sorption measurement with the sc-dried sample, activated under high vacuum at room temperature, had a similarly low N_2_ uptake (Appendix A).

Due to the low nitrogen uptake and small channel size of solvent-depleted [Zn(Isa-az-tmpz)], sorption experiments at 87 K for argon (Ar) and hydrogen (H_2_) and at 195 K for carbon dioxide were collected for the sc-dried and additionally heated material (3 h, 120 °C). While the Ar sorption measurement showed low gas uptake (Appendix A), the H_2_ sorption yielded an isotherm similar to type I(b) isotherms for microporous materials, with a total uptake of 2 mmol·g^–1^ (Figure 6) [47]. The CO_2_ sorption experiment showed the microporous nature of the material by also providing a type I(b) isotherm and a significant CO_2_ uptake, from which a Langmuir surface area of 588 m^2^·g^–1^ and a BET surface area of 496 m^2^·g^–1^ was calculated. Assuming the validity of the Gurvich rule, the division of (specific CO_2_ amount adsorbed in g g^–1^ with the CO_2_ saturation pressure at 195 K of 1.00 bar)/(density of liquid CO_2_ adsorbate with ρ_CO2_(195 K) = 1.08 g cm^–3^) gave a pore volume of 0.22 cm^3^ g^–1^ (the uptake of 120 cm^3^ g^–1^ at STP at 1 bar is 5.4 mmol g^–1^ or 2.4 g g^–1^) [48]. The difference in gas uptake between N_2_, Ar, H_2_ and CO_2_ correlates with the kinetic diameters (3.64, 3.40, 2.89 and 3.30 Å, respectively), with the cryogenic temperatures for N_2_ (77 K) and Ar (87 K) and the ultramicroporous (<7 Å pore size) nature of the framework. The diffusion of N_2_ molecules and Ar atoms into small pores was then very slow, while kinetic inhibition was less severe for smaller H_2_ molecules and for CO_2_ at 195 K.

## 3. Materials and Methods

### 3.1. Materials and Characterization

All reagents were obtained from commercial sources and used without further purification. C, H, N analyses were executed on a vario MICRO cube from Elementar Analysentechnik. ^1^H-NMR and ^13^C-NMR spectra were measured on a Bruker Avance III-300. IR-spectra were recorded on a Bruker Tensor 37 IR spectrometer equipped with an attenuated total reflection (ATR) unit (Platinum ATR-QL, Diamond). Electrospray ionization-mass spectra (ESI-MS) were measured on a Finnigan LCQ Deca Thermoquest in acetone, electron ionization-mass spectra (EI-MS) on a TSQ 7000 Finnigan MAT. Thermogravimetric analysis was executed on a Netzsch TG 209 F3 Tarsus in the range from 30 °C to 600 °C with a heating rate of 5 K min^–1^ under a nitrogen atmosphere. Powder X-ray diffraction (PXRD) patterns were measured on a Bruker D2 Phaser powder diffractometer with a flat silicon, low background sample holder, at 30 kV, 10 mA with Cu-Kα radiation (λ = 1.5418 Å). The most intense reflection in each diffractogram was normalized to 1. The simulated PXRD pattern has been calculated with MERCURY software [23]. Supercritical drying was carried out on a Leica EMPCD 300 over 99 exchange cycles with CO_2_. Adsorption data for N_2_ at 77 K (liquid nitrogen bath) was collected on a Quantachrome NOVA 4000 gas adsorption analyzer. Additional sorption experiments for Ar and H_2_ at 87 K (Quantachrome CRYOCOOLER) and CO_2_ at 195 K (Quantachrome CRYOCOOLER), respectively, were conducted on a Quantachrome Autosorb iQ MP. The supercritically dried sample was used and outgassed before the gas sorption measurements were taken, either at room temperature or at 120 °C for a minimum of 3 h.

### 3.2. Single Crystal X-ray Diffraction

Suitable crystals were carefully selected under a polarized-light microscope, covered in protective oil and mounted on a cryo-loop.

For crystals 1a and 2a, the data were collected on a Bruker Kappa APEX 2 CCD X-ray diffractometer with a microfocus sealed tube, Mo-Kα radiation (λ = 0.71073 Å), and a multi-layer mirror monochromator; data collection took place at 100 ± 2 K (crystal 1a) or 140 ± 2 (crystal 2a) using ω-scans, cell refinement with APEX2 [49], data reduction with SAINT [50] and experimental adsorption correction with SADABS [51].

For crystals 1b and 2b, the single crystal diffraction data were collected using a Rigaku XtaLAB Synergy S four circle diffractometer with a Hybrid Pixel Array Detector and a PhotonJet X-ray source for Cu-Kα radiation (λ = 1.54184 Å), with a multilayer mirror monochromator. Data collection took place at 100.0 ± 0.1 K using ω-scans. Data reduction and absorption correction were performed with CrysAlisPro 1.171.41.105a [52].

Structure analysis and refinement: The structures were solved by direct methods (SHELXT-2015), full-matrix least-squares refinements on *F*^2^ were executed using the SHELXL-2017/1 program package [53,54]. All hydrogen atoms were positioned geometrically (with C–H = 0.95 Å for aromatic CH and C–H = 0.98 Å for CH_3_) and refined using riding models (AFIX 43 and 137 with U_iso(H)_ = 1.2 U_eq_ (CH) and 1.5 U_eq_ (CH_3_)).

Highly disordered solvent molecules were either masked with the SQUEEZE option in PLATON [24,25] (crystals 1a and 2a) or by using the solvent mask feature as implemented in OLEX 2.1.3 [55] (crystals 1b and 2b). Crystal data and details on the structure refinement are provided in Table 1. Details about selected bond distances and angles are provided in Appendix A in the supporting information. Graphics were drawn with the program DIAMOND [56].

The crystallographic data (excluding structure factors) for the structures were deposited with the Cambridge Crystallographic Data Centre (CCDC-numbers 2192050-2192053) and can be obtained free of charge via www.ccdc.cam.ac.uk/data_request/cif.

### 3.3. Synthesis of H_2_Isa-az-tmpz

The ligand was synthesized following a modified previously reported procedure [13]. Dimethyl 5-(2-(2,4-dioxopentan-3-ylidene)hydrazineyl)isophthalate (2) was obtained via a Japp-Klingemann reaction of dimethyl 5-aminoisophthalate (1) with acetylacetone. Compound 2 was converted with methyl hydrazine into dimethyl 5-(2-(1,3,5-trimethyl-1H-pyrazol-4-yl)azo)isophthalate (3), which was later deprotected under basic conditions to obtain 5-(2-(1,3,5-trimethyl-1H-pyrazol-4-yl)azo)isophthalic acid (H_2_Isa-az-tmpz) (4) as a new ligand in 76% yield over three steps (Figure 2). The detailed synthetic procedure and analysis can be found in the Appendix A.

### 3.4. Synthesis of [Zn(Isa-az-tmpz)]·~1–1.5 DMF

In a Pyrex tube, 9.7 mg (0.032 mmol) of H_2_Isa-az-tmpz and 16.8 mg (0.064 mmol) of Zn(NO_3_)_2_·4 H_2_O were dissolved in 2 mL of DMF. The tube with the yellow solution was placed into a preheated oven at 80 °C for 72 h to obtain yellow block-shaped crystals. The mother liquor was then exchanged against fresh DMF (2 mL) to prevent further growth and stored at RT until single crystal analysis.

On a larger scale, 100 mg (0.33 mmol) of H_2_Isa-az-tmpz dissolved in 10 mL of DMF was added to a solution of 170 mg (0.65 mmol) Zn(NO_3_)_2_·4 H_2_O in 10 mL of DMF. The same temperature program used for the single crystal synthesis was applied. The obtained as-synthesized polycrystalline yellow material was washed with DMF (10 mL). The solvent was replaced once a day for three days followed by acetone (10 mL), which also had the solvent replaced once a day for three days. Subsequently, the sample was activated via supercritical drying with CO_2_ yielding 81.4 mg (67 %) of a yellow precipitate as the activated material.

Calculated for as-synthesized [Zn(Isa-az-tmpz)]·1 DMF (ZnC_17_H_19_N_5_O_5_), C 46.54%, H 4.37%, N 15.96%; found C 46.36%, H 4.37%, N 15.66%;

Calculated for activated [Zn(Isa-az-tmpz)] (no DMF) (ZnC_14_H_14_N_4_O_4_), C 45.99%, H 3.31%, N 15.32%; found for sample 1: C 45.95%, H 4.37%, N 14.97%; sample 2: C 46.60%, H 3.72%, N 14.56%.

IR for activated [Zn(Isa-az-tmpz)] [cm^–1^]: 1677 (w), 1648 (s, COO^–^), 1560 (w), 1521 (w), 1438 (m, CH_3_), 1372 (s, C-N), 1254 (w), 1233 (w), 1091 (w), 1061 (w), 1040 (w), 1008 (w), 974 (w), 921 (w), 865 (w), 805 (w), 778 (m, Ar-H), 717 (m, Ar-H), 682 (m, Ar-H), 635 (w), 606 (w), 588 (w).

## 4. Conclusions

We presented the synthesis of the MOF [Zn(Isa-az-tmpz)]·~1–1.5 DMF with the rare chiral **3,6T22**-topology through the reaction of zinc nitrate with the newly synthesized T-shaped linker 5-(2-(1,3,5-trimethyl-1H-pyrazol-4-yl)azo)isophthalic acid (H_2_Isa-az-tmpz) in DMF at elevated temperatures. Even though the deprotonated form Isa-az-tmpz^2–^ fulfills the general principles of the ‘ligand-to-axial pillaring’ strategy of the supramolecular building layer (SBL) approach, leading to a 3,6-c connected MOF, the resulting topology does not follow the SBL approach and cannot be described through this. Contrary to the related coordination polymer **rtl**-[Zn(HIsa-az-dmpz)], which could not be activated to its porous material, the porosity for the ultramicroporous MOF (pore diameters less than 7 Å) **3,6T22**-[Zn(Isa-az-tmpz)]·~1–1.5 DMF could be assessed with H_2_ at 87 K and with CO_2_ at 195 K but not with N_2_ and Ar under the cryogenic conditions of 77 and 87 K, respectively. Overall, this work can be considered a starting point to obtain chiral and potentially porous MOFs using bifunctional and T-shaped pyrazole-carboxylate ligands. Chiral MOFs are promising materials for enantioselective adsorption, separation and catalysis [20,21,22].

## Data Availability

The data presented in this study are available on request from the corresponding author.

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
