# Peer review of "Synthesis of a Chiral 3,6T22-Zn-MOF with a T-Shaped Bifunctional Pyrazole-Isophthalate Ligand Following the Principles of the Supramolecular Building Layer Approach"

_molecules, 2022, doi:10.3390/molecules27175374_

Round 1
Reviewer 1 Report
Review on the paper entitled:
"Synthesis of a chiral 3,6T22-Zn-MOF with a T-shaped bifunctional pyrazoleisophthalate ligand following the principles of the supramolecular building layer approach."
In this paper, the authors describe the design and synthesis of Zn-based chiral MOFs with a T-shaped ligand. They test the gas-sorption properties of the obtained MOF after different activation conditions.
The study is very well done and scientifically correct. The results are presented logically and solidly. The manuscript itself is well-written, with attention to detail.
However, in my opinion, a couple of points would definitely improve the manuscript.
Some comments.
1) The authors should put their work more in perspective, comparing the internal structure of the MOF to the analogs in the literature.
2) Solid-state chirality on early Earth is (possibly) at the origin of biological chirality. That is why chiral MOFs could be of great interest. For instance, in catalysis. Maybe the authors could comment on this point in the introduction/conclusions?
3) Azo-ligand might have the potential for photoisomerization. Do the authors consider the possibility of utilizing this property to use light for post-synthetic photo-isomerization of the described COF?
Some questions.
1) On the molecular formula of MOF: [Zn(Isa-az-tmpz)]·xDMF – why do you use x for the quantity of DMF molecules? In the experimental part, the elemental analysis is given for MOF with 1 DMF molecule.
2) Figure 5. The authors mention that upon exchange of DMF for acetone, no phase change was observed with PXRD. However, it looks like some changes happened. How do the authors explain the changes in intensities for bands at ca. 10° 2θ, for instance?
Minor corrections.
1) The first line in the Introduction: Metal-organic frameworks (MOFs) are is a much-studied topic…
2) Which factors influence the “structure design of MOFs” except the imagination of the chemist? The point here is of the “design”, therefore there are not many factors influencing it objectively.
3) Homogenize the structure of Zn(NO3)2•4H2O through the text. In some cases, it is with a space; in some – without.
4) Add letters a), b) c), etc. in Figure 5, and cite them in the text correspondingly.
5) Homogenize the values of dimensions. Either one uses “g/cm3” or “g cm–3”. One should avoid mixing the styles.
6) The minor point to improve would also be the conclusions, which should frame more the actual discovery in this paper, and, possibly, frame the playground for the follow-up studies or/and for their use in the development and understanding of similar reactions.
Overall, the present paper is well suited to be published in Molecules.

Author Response
Please see the detailed answers in the attached file.

Reviewer 2 Report
This paper includes the preparation of a chiral 3,6T22-Zn-MOF with a T-shaped bifunctional pyrazole isophthalate ligand. Its crystal structure has been determined and shows an interesting novel MOF
Crystal structures are obtained of four samples, all showing exactly the same structure. It is not at all clear why the authors have included 3 of them in the paper. The authors do not discuss any differences in the four samples (so it can be concluded that there aren’t any). All they offer is different Flack parameters. There is no good reason to include crystallographic details of all four samples in Table 1.
However if the authors wish to include details of structures 1b, 2a or 2b in the paper or indeed to submit them to the CCDC, then because of the isomorphism, it is necessary to include the same coordinates for each structure. For example 1a has the zinc in 0.59, 0.75, 0.48 and 1b is 0.24, 0.59, 023 which is inconsistent. Symmetry transformations should be applied so that the packing diagrams apply to all four structures. Though In my view the only reason for including structures 1b, 2a, 2b in this paper is to quote the Flack parameters which are of interest.
The structures have been well determined and there are no errors in the cif files.. Though it is necessary to include the missing terms for_exptl_absorpt_correction_T_max and _exptl_absorpt_correction_T_min which are omitted.
I suggest replacing the term absolute structure parameter with the more familiar Flack parameter in Table 1
However the structure of 1a has been successfully refined and the description of the structure is clearly written and the diagrams are good
In the Supplementary Material, it should be made clear that all details of the structure and calculations refer to structure 1a.
There follows gas sorption studies which are of interest.
I recommend the paper be accepted after minor revision
Author Response

(The authors gave the same response as above.)

Reviewer 3 Report
The authors describe the synthesis and characterization of a MOF with T-shaped linker.
The crystals obtained belong the an enantiomeric space group P43212 or its inverse P41212.
As the crystals are not enantiomeric pure (TWIN/BASF) refinement the term 'racemic conglomerate' is NOT to be used as according to the iupac definition this term is reserved for enantio-pure crystals
[An equimolar mechanical mixture of crystals each one of which contains only one of the two enantiomers present in a racemate . The process of its formation on crystallization of a racemate is called spontaneous resolution, since pure or nearly pure enantiomers can often be obtained from the conglomerate by sorting.]
The term racemic twin is also not appropriate here:
On p4 the phrase "The four investigated crystals (Table 1) were refined as racemic twins with ratios ..."
should read "The four investigated crystals (Table 1) were refined as twinned by inversion with ratios ..."
(the term racemic twin being the ideal case with a 50/50 ratio)
In section 2: I don't see the added value of showing the morphology of the crystal. if this were to be reported, the face indexing tool in apex2 or crysAlisPRO should be used to get the actual faces.
Author Response

(The authors gave the same response as above.)
